# A Tale of Ice and Fire: The Dual Role for 17β-Estradiol in Balancing DNA Damage and Genome Integrity

**DOI:** 10.3390/cancers13071583

**Published:** 2021-03-30

**Authors:** Sara Pescatori, Francesco Berardinelli, Jacopo Albanesi, Paolo Ascenzi, Maria Marino, Antonio Antoccia, Alessandra di Masi, Filippo Acconcia

**Affiliations:** 1Department of Sciences, Section Biomedical Sciences, and Technology, University Roma Tre, Viale Guglielmo Marconi, 446, I-00146 Rome, Italy; sara.pescatori@uniroma3.it (S.P.); francesco.berardinelli@uniroma3.it (F.B.); jacopo.albanesi@uniroma3.it (J.A.); paolo.ascenzi@uniroma3.it (P.A.); maria.marino@uniroma3.it (M.M.); 2Neurodevelopment, Neurogenetics and Molecular Neurobiology Unit, IRCCS Santa Lucia Foundation, 00143 Rome, Italy; 3Neuroendocrinology, Metabolism and Neuropharmacology Unit, IRCCS Santa Lucia Foundation, 00143 Rome, Italy

**Keywords:** 17β-estradiol, estrogen receptor alpha, breast cancer, DNA damage, DNA repair

## Abstract

**Simple Summary:**

Paradoxically, although the steroid hormone 17β-estradiol (E2) regulates many aspects of male and female physiology, it is described in the chemical indexes as a carcinogen. By the analysis of the literature, we unveil here a novel concept for which E2 possesses a dual nature for which it is both a toxic and a homeostatic regulator controlling DNA stability. Therefore, cancer could arise as a consequence of the deregulation of this delicate equilibrium between cellular pathways.

**Abstract:**

17β-estradiol (E2) regulates human physiology both in females and in males. At the same time, E2 acts as a genotoxic substance as it could induce DNA damages, causing the initiation of cellular transformation. Indeed, increased E2 plasma levels are a risk factor for the development of several types of cancers including breast cancer. This paradoxical identity of E2 undermines the foundations of the physiological definition of “hormone” as E2 works both as a homeostatic regulator of body functions and as a genotoxic compound. Here, (i) the molecular circuitries underlying this double face of E2 are reviewed, and (ii) a possible framework to reconcile the intrinsic discrepancies of the E2 function is reported. Indeed, E2 is a regulator of the DNA damage response, which this hormone exploits to calibrate its genotoxicity with its physiological effects. Accordingly, the genes required to maintain genome integrity belong to the E2-controlled cellular signaling network and are essential for the appearance of the E2-induced cellular effects. This concept requires an “upgrade” to the vision of E2 as a “genotoxic hormone”, which balances physiological and detrimental pathways to guarantee human body homeostasis. Deregulation of this equilibrium between cellular pathways would determine the E2 pathological effects.

## 1. Introduction

The sex hormone 17β-estradiol (E2) exerts diverse pleiotropic physiological effects including the control of the reproductive system in females and the development of primary and secondary sexual characteristics in humans. E2 regulates a plethora of physiological functions in non-reproductive tissues including heart, bone, and brain systems. Accordingly, E2 can exert beneficial effects being protective against osteoporosis, cardiovascular and neurodegenerative diseases [1].

The E2 effects occur mainly as the result of hormone binding to estrogen receptor subtypes α and β (i.e., ERα and ERβ), which display different patterns of expression in tissues although the two receptors show a high degree of sequence homology. The E2 molecular action through ERα and/or ERβ justifies the diverse and sometimes contrasting effects of this hormone. As an example, E2 can both induce or inhibit cell proliferation by binding to ERα or ERβ, respectively [1].

Thus, it is not surprising that E2 is involved in the growth and survival of several types of human cancers including breast cancer (BCs) [2]. Although BC is a heterogeneous disease with different molecular phenotypes, the most frequent BCs (i.e., 75%) express the ERα at the diagnosis. The ERα is an important prognostic factor because its expression drives the treatment (i.e., the endocrine therapy), which aims to block different aspects of the E2:ERα proliferative signaling [3].

Conversely, the increased plasma level of E2 is a well-defined risk factor for BC [4,5,6]. Although the mechanisms through which this hormone determines its detrimental effects are not completely understood and imply also its interplay with other hormones (e.g., progestins) [4,5,6], E2 has been shown to cause directly or indirectly DNA damage [7]. E2 could contribute to the initiation and progression of BC through the induction of DNA double-strand breaks (DSBs) and genomic instability [8,9,10]. Indeed, E2 can induce the production of oxidative metabolites, which cause DNA adducts and/or oxidative DNA damage [9]. Also, the hyperactivated E2:ERα signaling provokes excessive proliferation in turn promoting DNA damage accumulation (e.g., replication fork stalling-dependent DSBs) [11,12].

To face DNA damage, cells have evolved a complex set of mechanisms termed DNA damage response (DDR), which allows the detection of the lesions, activates the damage signaling, and regulates the DNA repair [13,14]. It has been suggested that E2 signaling inhibits the DDR to induce chromosomal instability and aneuploidy, which are typical cytogenetic prerequisites for BC initiation and progression [7]. Moreover, E2 induces a physiological transcriptional and replicative stress (RS) on nucleic acids [15]. Remarkably, RS commonly occurs in cells and causes DNA damage, which is rapidly counteracted by the RS response (RSR) [16]. Therefore, this evidence suggests a complex E2-dependent modulation of the cellular pathways controlling both DDR and RSR for the fine regulation of the E2-induced cell proliferation. Nonetheless, it is difficult to reconcile how E2, which controls crucial physiological processes both in females and males and contributes to body homeostasis, could work as a carcinogen inducing the development of BC.

The mechanisms by which E2 elicits DNA damage and protect cells from genomic instability as well as the ability of this hormone to directly control the signaling underlying DDR and RSR are reviewed here. Reported evidence allows proposing that E2 balances DNA damage and genome stability via an intertwined cross-talk involving ERα, DDR, and RSR signaling.

## 2. The Molecular Pathways of E2:ERα Signaling to Cell Proliferation

E2 induces cell proliferation by activating both nuclear and extra-nuclear ERα activities (Figure 1).

ERα is a ligand-activated transcription factor belonging to the nuclear receptor superfamily. E2 binding to ERα induces E2-dependent gene transcription by triggering ERα association to the promoters containing the estrogen-responsive-element (ERE) sequence. The association of specific transcriptional co-activators and co-repressors to ERα contributes to the modulation of gene transcription (Figure 1). ERα can also modulate genes that do not possess the ERE-sequence in their promoters because the hormone-bound receptor can associate with other transcription factors (e.g., AP-1, SP-1, and NF-kB), which, in turn, expand the repertoire of the E2-regulated genes [1]. The E2:ERα transcriptional activity is further controlled by the membrane-starting signaling activated by E2 [2]. ERα is also located at the plasma membrane through lipid modification (i.e., palmitoylation) mediating, upon E2 binding, the activation of a plethora of signaling cascades (e.g., ERK/MAPK; PI3K/AKT pathways) [17,18,19] (Figure 1). In mice models harboring the mutation of ERα palmitoylation site (i.e., C447 to A in human ERα and C451 to A in murine ERα) it has been firmly demonstrated that the extra-nuclear plasma-membrane-dependent signaling of the E2:ERα plays paramount roles in the regulation of the physiological effects of E2 in vivo [20,21] as they intimately integrate with the transcriptional functions of the E2:ERα complex [18].

The activation of the above-mentioned E2-induced cellular signaling ultimately results in the induction of DNA synthesis, which allows cell cycle progression, cell duplication, and cell proliferation (for extensive review, please see for example [22,23]).

### Context-Dependent Effects of E2

The pleiotropic nature of E2 implies that this hormone can produce diverse effects in different cellular contexts. Although the specific molecular details linking the repertoire of E2-dependent effects with the structure-function relationships played by the ERs within the cellular environments are still not completely understood, numerous evidence can explain the context-dependent effects of E2.

The ERα and the ERβ are differentially expressed in different tissues. Indeed, the ERs can be singularly expressed in one tissue or co-expressed at different levels in a specific cellular context and their intracellular levels can vary as a function of time and tissue physiological state [24].

Moreover, at the cellular level, it is well known that the ERα and the ERβ possess a high number of interacting partners [25,26,27]: the complexity of the ERs interactomes is not only correlated to the regulation of the expression of the ERs interacting partners in the different cellular contexts but also to the biochemical properties of the ERs. The ERs are modular proteins, which contain only two folded domains (i.e., the DNA binding domain -C domain; the ligand-binding domain: E domain) out of six (i.e., A to F) [28,29]. These two structured parts of the proteins are then linked to the other domains through intrinsically disordered regions (IDRs). IDRs confer to proteins a high degree of dynamic flexibility, increase the possibility that the same protein could interact with various partners, allow that the same protein could contribute to the regulation of many intracellular pathways, and are the target of post-translational modifications (PTMs) [30]. The ERα and the ERβ appear to have acquired all the properties of the IDR-containing proteins: the ERα and the ERβ expose different epitopes in different subcellular compartments [31]; they interact with different binding partners both in different subcellular compartments and in different tissue contexts [25,26,27], in this way contributing to the regulation of many different intracellular pathways [1,2]; the ERα and the ERβ are the targets of a plethora of PTMs [32] and the IDRs of the ERα can assume a different folding as a function of the specific receptor binding partner [33].

Furthermore, the complexity of the ERs signaling is increased by the fact that (i) E2 plasma levels physiologically fluctuate as a function of time (see below), (ii) more than one ligand-binding site has been suggested to be present within the ERs structure [34,35] and (iii) E2 elicits complex transcriptional and non-transcriptional cellular responses [1,2].

Therefore, given the high heterogeneity of the functional panorama of E2 effects and ERs structures as well as of the molecular actors mediating the effects of the E2-regulated pathways in different cell types, in this work, the mammary gland cells are the main context where the effects of E2 on the balance among DNA damage, protection from genomic instability and control of DDR and RSR signaling are reported.

## 3. Relationships between E2 Concentrations and E2:ERα Signaling to Cell Proliferation

One possibility through which E2 can be at the edge between a hormone or a genotoxic compound could be its concentration-dependent effect. In fertile women, E2 plasma levels physiologically fluctuate (from low pM to 1 nM at the ovulation peak), thus regulating female fertility [36]. In addition, the increase in plasma levels is a risk factor for the insurgence of BC [4,5,6]. Therefore, one might expect deep knowledge of the E2 concentration-dependent effects on the E2:ERα signaling to cell proliferation, but instead, a systematic analysis addressing this point is still lacking.

Nonetheless, it is well known that in vitro the concentration-dependent effects of E2 on both ERα-triggered ERE-based gene transcription and cell proliferation usually follow a bell-shaped curve with low- and supra-physiological concentrations of E2 (i.e., pM and μM, respectively) having the same effects. Remarkably, these observations have been also confirmed in animal models where different tissues are differentially sensitive to low or high levels of E2 [37].

In this respect, E2-induced gene transcription requires the ability of ERα to cycle on-and-off its target gene promoters. This fluctuation of receptor binding to chromatin occurs in an ordered way. Thus, the E2:ERα complex could sequentially recruit both co-activators and proteins of the basal transcriptional apparatus (e.g., RNA Pol II) to coordinate gene expression with the fluctuating plasma concentration of E2. The regulation of this intricate network is controlled by E2 itself, which induces ERα degradation to limit the excessive response to the hormone stimulation [38,39]. Genome-wide chromatin immunoprecipitation sequencing (CHIP-seq) confirms that ERα interacts with millions of chromatin binding sites with or without E2 [40,41,42], thus controlling the genomic landscape of the E2-target cells.

Interestingly, although the E2 concentration-dependent modulation of these molecular circuitries has not been studied, real-time live-cell analyses of E2:ERα-mediated ERE-based transcription demonstrated that the activation of the transcriptional activity dependent on the E2:ERα complexes is rapid and detectable as early as 3 h after administration of 10 pM E2 [43]. E2-dependent transcriptional activity occurs with an effective dose 50 (ED_50_) of ~5 pM at 24 h (Figure 2a) [43]. The gene transcription then induces the activation of processes required for DNA synthesis. In addition, in this case, the regulation of DNA synthesis induced by different E2 concentrations has not been thoroughly investigated. Recent evidence in mice shows that supraphysiological concentrations of E2 inhibit uterine epithelial cell proliferation and, consequently, the E2-induced increase of uterine weight [37]. In ERα overexpressing cells, E2-induced DNA synthesis occurs as rapid as 6 h [44,45]. In BC cells endogenously expressing the ERα, the DNA synthesis occurs 24 h after the administration of ~20 pM E2 (Figure 2b). Thus, picomolar concentrations of E2 are sufficient to activate via ERα intense waves of both gene transcription and DNA synthesis.

Because nanomolar concentrations of E2 at the ovulation peak (i.e., 1 nM) activate E2:ERα complex-dependent gene expression and DNA duplication at maximal levels and supra-physiological concentrations of E2 can inhibit cell proliferation [37], high concentrations of E2 could be detrimental for cellular and tissue functions by damaging the genome through physical breaks on the DNA helix (please see below).

## 4. Molecular Pathways for Genome Stability Maintenance

The DNA damage is recognized and processed by spatially and temporally controlled pathways that are collectively called DNA damage response (DDR) (Figure 3).

DDR is required: (i) to detect the DNA lesion; (ii) to guide the correct damage signaling, (iii) to coordinate transcriptional and post-translational activation of genes involved in the DNA repair; (iv) to promote the activation of the cell cycle checkpoint; and (v) to eventually induce apoptosis or senescence to deplete or to semi-permanently arrest irreversibly damaged cells [7,46,47]. Of note, these pathways are activated also during the RSR. Under pathological conditions, such as cancer, the regulation of DDR is impaired or abrogated and the risk for genome instability is increased due to the lack of the proper DNA repair mechanism.

ATM, ATR, and the catalytic subunit of DNA-PKcs, which associates with the Ku70/80 heterodimer (Ku) to form the DNA-PK holoenzyme [48], are large Ser/Thr kinases and members of the phosphatidylinositol 3-kinase-related kinase (PIKK) family. These kinases are central regulators of DDR that activate, through redundant mechanisms, three possible signaling axes, i.e., the ATM-CHK2, the ATR-CHK1, and the DNA-PK cascades [48,49,50,51,52,53] (Figure 3).

ATM kinase is recruited and activated in the presence of a DSB thanks to the MRE11/RAD50/NBS1 (MRN) sensor complex [54,55,56]. ATR is activated together with its partner ATRIP in response to a single-strand break (SSB) thanks to the presence of the replication protein A (RPA) sensor [57,58,59,60,61]. DNA-PKcs is activated by DSB and is activated by Ku [48,62]. Once activated, ATM, ATR, and DNA-PK phosphorylate H2AX (i.e., the so-called γH2AX) and hundreds of mediator proteins (Figure 3).

ATM and ATR trigger also a second wave of phosphorylation through the activation of CHK2 and CHK1 protein kinases, respectively [63,64]. The ATM-CHK2 and ATR-CHK1 axes also stimulate the transcriptional activity of p53, which promotes a temporary arrest in the cell cycle, thus allowing DNA repair and restoration of homeostasis [51]. Once the damage is repaired, all these post-translational modifications are reversed to stop the DDR signal [13,65,66,67,68] (Figure 3).

## 5. The Interplay among E2, E2:ERα Signaling, and Genome Stability Maintenance

Notwithstanding the deep knowledge regarding the mechanisms through which the E2:ERα controls gene expression and DNA duplication, only limited, scattered and uneven information is available about the impact of E2 and/or E2:ERα signaling on the regulation of both DNA damage and DNA damage response and repair.

### 5.1. E2 as a Source of DNA Damage

Many in vitro and in vivo studies have highlighted the ability of E2 to induce DNA damage (Figure 4 and Figure 5). E2 appears to induce DNA lesions by both its chemical properties (i.e., acting by a direct carcinogen independently on cell cycle) and its functional activity (i.e., inducing RS and therefore, restricted to S-phase). In humans and rodents, the E2 administration increases the incidence of mammary, ovarian, colon, prostate, and endometrial tumors [4,5,6,8], thus it appears not to be context-dependent. The role of E2 in carcinogenesis occurs through the pro-carcinogen metabolic activation (i.e., genotoxic “initiator”) and the E2:ERα complex activation (i.e., “promoter”) [37,40,41,42,43,44,45]. The two mechanisms of mammary carcinogenesis are not necessarily mutually exclusive and may even concomitantly act [69].

#### 5.1.1. E2 as a Direct Carcinogen

E2 is metabolized by phase I P450 enzymes in the liver, uterus, mammary gland, and testis to generate catechol estrogens (CE) due to the oxidation at C-16 (16α-OHE2), C-2 (2-OHE2), and C-4 (4-OHE2). Their relative yield depends on tissue metabolism. In breast tissues, CYP1A1 and CYP1B1 are mainly involved in the E2 conversion and, interestingly, the enzyme levels are regulated by E2 via ERα [70]. Urinary levels of 2-OHE2 and 4-OHE2 are elevated in BC patients compared to healthy controls [71] and elevated concentrations of 4-OHE2 have been detected in BC biopsies [72]. In turn, hydroxyestradiols, particularly 4-OHE2, undergo CYP1B1-mediated one electron-redox cycle to semiquinone (SQ) intermediate and ortho-quinones derivatives (Catechol Estrogen Quinones, CEQs). The SQs electrophilic metabolites E2-3-4-Q and at a lesser extent E2-2-3-Q are known to attack DNA and form depurinating adducts at the N7-position of guanine and the N3-position of adenine (4-OHE2-1-N7Gua, 4-OHE2-1-N3Ade) at picomolar concentrations, whereas the lesser reactive E2-2,3-Q yields 2-OHE2-6-N3A adducts (Figure 4). Such DNA adducts have been associated with increased cancer risk and appear ultimately to act as carcinogenic metabolites. Therefore, apurinic sites are potentially mutagenic if not faithfully repaired [73,74,75] (Figure 4).

In contrast, with hydroxyestradiols, the inability of E2 to damage isolated calf thymus DNA demonstrates that its metabolic conversion is a prerequisite to act as a DNA-damaging agent. CE oxidation to quinones is maintained in homeostasis by phase II enzymes, thus minimizing the production of reactive species. In extrahepatic tissues, CE is inactivated by catechol-O-methyltransferase (COMT) [76]. Of note, the COMT inhibitor Ro41-0960 impairs metoxilathion of catechol estrogens leading to a significant increase of depurinating DNA adducts [77]. However, such a protective mechanism is reduced by the capability of E2 to act as a COMT down-regulator inhibiting its promoter DNA methylation [78]. The level of circulating CEQs is also regulated by the catalytic action of glutathione *S*-transferase GSTP1, which conjugates CEQs with glutathione (GSH) [79]. Also, plant-derived polyphenols (e.g., resveratrol) may act as protective agents against E2-metabolites. Resveratrol has antioxidant activity, positively modulates phase II enzymes, and efficiently counteracts E2-DNA adducts formation and neoplastic transformation of cultured normal epithelial breast MCF-10F cells [80,81].

The E2-induced-DNA damage is not restricted to the above-mentioned DNA-adducts. Indeed, reactive oxygen species (ROS) produced during E2 metabolization are potent inducers of oxidized bases (e.g., 8-hydroxy-2’-deoxyguanosine (8-OHdG), 5-hydroxymethyl-2’-deoxyuridine (HMdU)) inducing DNA single-strand breaks (SSBs) (Figure 4). E2 metabolites can also oxidize proteins and lipids [8]. In particular, poly-unsaturated lipids are easily peroxidized and may give rise to aldehyde DNA adducts [82]. The cellular antioxidant status following E2 exposure is modulated by ERα. MCF-7 cell treatment with 10 nM E2 leads to a significant reduction of cell ability to metabolize peroxide. This reduction reflects the decrease of catalase activity and glutathione levels as well as the increased peroxide-induced DNA damage. In this experimental setting, E2 increased levels of glutathione peroxide, SOD1 and SOD2 [83]. Recently, it has been reported that E2 is also able to increase mitochondrial ROS production in MCF-7C cells [84].

Induction of SSBs, alkali labile sites, and oxidized purines has been documented by COMET assay in both ERα-positive (MCF-7) and ERα-negative (MDA-MB-231) cells exposed to E2 (10–1000 nM) and 4-OHE2 (4–100 nM) [85]. However, it should be considered that the concentration of E2 used in this study exceeded physiological levels of the hormone occurring in healthy women (1 nM) [85]. Nonetheless, in a previous publication [86], it was demonstrated by COMET assay that 0.1 nM of E2 induced DNA damage and micronucleus formation in MCF-7 cells.

E2 and E2 metabolites can also cause DNA DSBs in human ERα-positive and -negative cells. Ten nanomolar E2 causes DSBs in proliferating ERα-negative epithelial MCF-10A cells, in which BRCA1 plays a major role in DNA repair to prevent genomic instability [74]. Interestingly, BRCA1 acts also as a repressor of CYP1A1 transcription. E2-induced DSBs do not only occur exclusively in the S-phase cells via a possible transcription-collision mechanism but also in G0/G1-phase cells. This ERα-dependent mechanism observed in MCF-7 cells [87] relies on the formation of abortive TOP2B catalysis generating pathological stalled TOP2-adducts at transcriptional regulatory sequences (promoters and enhancers). If adducts are not properly repaired by BRCA1, BRCA2, Mre11, and by the non-homologous end joining (NHEJ) pathway, unrepaired DSBs occur [88]. Therefore, the mechanism for DSBs formation after E2 exposure may be dependent on replication-transcription collision during S-phase or TOP2B-adducts throughout the cell cycle (see below) [88].

Therefore, E2 can induce DNA damage (i.e., DNA adducts, SSBs, DSBs) through the hormone cellular metabolism, which determines either the production of intermediates potentially causing damage to the DNA or the increase in ROS, which can be, directly and indirectly, detrimental for genome integrity (Figure 4). However, under conditions where the catechol pathway is balanced by the set of the above-mentioned protective enzymes, the formation of ultimate carcinogenic metabolites of E2 is minimized; furthermore, because CEQs are rapidly cleared by the liver and kidneys their half-life is relatively low. Nonetheless, mutations in the enzymes that control E2 metabolism have been shown to be a carcinogen [89,90]. Therefore, E2 mainly acts as an epigenetic carcinogen by stimulating abnormal cell proliferation via the engagement of the E2:ERα-mediated pathways [75].

The DNA damaging effect of E2 can occur independently on ERα via the metabolic-produced carcinogens, as well as through the signaling activity of the E2:ERα complex. Thus, both mechanisms can coexist and can contribute to E2-mediated carcinogenesis.

#### 5.1.2. E2 as a Source of Replicative Stress

RS is characterized by DNA synthesis slow down and/or replication fork stalling/collapse and is triggered by many endogenous or exogenous events, which interfere with DNA replication and hamper its progression.

In the early stages of tumorigenesis, genomic instability occurs because of oncogene-induced RS. As outlined above, the E2:ERα-mediated pathways induce a dramatic increase in transcription and DNA synthesis, all these conditions possibly contributing to the E2-dependent induction of RS [91,92,93,94]. Accordingly, one model proposed to explain E2-induced genome instability suggests that the unrestrained proliferation driven by deregulation of genes such as cyclin D1 causes RS and DNA damage [7,11].

A second model that could explain RS in ERα-positive BC cells implies that the E2-induced transcriptional burst can contribute to RS and genome instability through the E2-dependent increase of co-transcriptional structures formed by RNA-DNA hybrids. These structures consist of nascent transcript hybridized to template DNA and are named R-loops [15]. They are frequently observed in mammalian genomes and are thought to play regulatory roles influencing the chromatin architecture of gene promoters and facilitating the transcription termination [95,96]. E2 treatment rapidly induces R-loops mostly at E2-responsive genes, but DNA damage arises only when cells enter in the S-phase, indicating that RNA-DNA hybrids hinder replication fork progression. This supports the association between R-loop-dependent DNA damage and DNA replication [15]. Interestingly, it has also been demonstrated that exposure to E2 induces γH2AX foci, a well-known marker of DSBs, in ERα expressing BC cells and in an S-phase-dependent manner [10]. The formation of γH2AX foci requires ERα and TOP2B and is inhibited by the DNA polymerase inhibitor aphidicolin [10], thus indicating a direct correlation between DNA damage and replication. Furthermore, γH2AX foci colocalize with Rad51, suggesting that the homologous recombination (HR) repair pathway faces E2-induced DSBs. Accordingly, E2-dependent ATR downregulation does not completely turn off the signaling involved in the RSR [10] (Figure 5).

These processes appear to be general as even male sex hormone androgen can induce DNA damage in prostate cells. This androgen-induced DNA damage can represent the mechanism producing specific genomic rearrangements typical of prostate cancer. Indeed, androgen signaling in neoplastic prostate cells induces TOP2B-mediated DSBs at many genomic loci. Therefore, it is possible that multiple different enzymatic activities, including that of TOP2B in the case of androgen signaling and that of other nucleases in the case of high levels of exogenous genotoxic stress, can cooperate with androgen signaling to generate recombinogenic DSB, the most frequent rearrangements found in prostate cancers [97].

Therefore, E2 can promote the formation of DNA lesions through the induction of transcription and DNA synthesis, which represent perturbating conditions of the replicative process. Moreover, the reported evidence together with the fact that also androgen determines effects like those elicited by E2 strongly indicates that sex steroid hormones can induce genome instability by interfering with the RSR repair systems.

### 5.2. The E2-Dependent Regulation of DDR and RSR Signaling

Direct links exist between the E2:ERα signaling and the cellular DDR and RSR signaling (Figure 5).

#### 5.2.1. The E2-Dependent Control of the DNA Damage Response Signaling

Because E2 can directly or indirectly cause DNA damage, one might expect that E2 would act as a hormone inducing the activation of the signaling pathways required to repair DNA lesions.

Recent data have shown that ATM is negatively regulated by the E2:ERα complex [7,10] through the upregulation of miRNA 18a and 106a expression, as demonstrated by cellular models and clinical samples of BC [10] (Figure 5). This regulation, besides other mechanisms, could explain the resistance of ERα-negative BC to chemotherapy and radiotherapy. Indeed, the high levels of ATM in ERα-negative BC suggest that this DDR kinase could represent an interesting drug target in ERα-negative BCs: the usage of specific ATM inhibitors in combination with chemotherapeutic agents and/or radiotherapy might achieve more effective clinical benefits as the treatment might enhance tumor sensitivity to both chemotherapeutics and radiotherapy [10].

DNA-PK plays a central role in RNA polymerase-dependent transcription [98]. The catalytic subunit Ku70/Ku80 of DNA-PK (DNA-PKcs) possesses a high affinity for DNA ends and rapidly interacts with DNA after DSBs induction [99]. Of note, E2 promotes ERα binding to Ku70, which contributes to the transcriptional functions of the receptor [100] (Figure 5). DNA-PK phosphorylates the Ser118 residue located in ERα, thus, promoting receptor stabilization and full transcriptional activity [100]. Noteworthy, not only ERα is a target of DNA-PKcs but also DNA-PK is a target of ERα. Indeed, two ERα-binding sites in a region upstream of the DNA-PKcs transcriptional initiation site are necessary for the E2:ERα-dependent regulation of the DNA-PKcs levels [101]. Physiologically, it has been proposed that DNA-PK could limit excess ERα degradation to balance the cellular response to E2 stimulation. However, during pathophysiological conditions accompanied by excessive E2 stimulation or during irradiation, this delicate balance can be altered [100]. Therefore, although it has been shown that E2:ERα negatively regulates ATM, these findings raise the possibility that E2:ERα signaling could sustain proliferation by promoting DNA-PK-mediated NHEJ to maintain genome integrity, rather than engaging the ATM-dependent high-fidelity DNA repair mechanisms [100]. In this way, E2 could activate the DDR pathway to protect the genome.

In addition, cyclin D1 is recruited by E2 to contribute to the regulation of the DDR pathway via an extra-nuclear mechanism [61]. Thus, ERα-cyclin D1 binding at the cytoplasmic membrane augments AKT phosphorylation (Ser473) and γH2AX foci formation. In the nucleus, cyclin D1 enhances homology-directed high-fidelity DNA repair [61]. Cyclin D1 is also recruited to γH2AX foci by E2 and induces Rad51 expression [10]. Moreover, E2:ERα signaling antagonizes the anti-proliferative and pro-apoptotic DDR signals in tumors. Thus, ERα signaling can sustain proliferation in situations where otherwise DNA damage would induce a cell cycle arrest and apoptosis [7].

#### 5.2.2. The E2-Dependent Control of the Replicative Stress Response Signaling

RS activates a surveillance pathway known as the replication checkpoint [102] that ensures replication completion and prevents replication fork breakage. ATR is the central kinase of the replication checkpoint pathway (Figure 3). ATR and its partner ATRIP are recruited to stalled replication fork by the accumulation of RPA on single-stranded DNA. Activation of ATR depends on TOPBP1, a protein recruited at single and double-stranded DNA junctions by the 9-1-1 complex (i.e., RAD1, RAD9, and HUS1) and its clamp loader RFCRAD178. Active ATR phosphorylates the effector kinase CHK1 at Ser317 and Ser345, a process also mediated by clapsin, Timeless (TIM), and TIPIN [103,104,105]. When fully activated, the ATR-CHK1 pathway regulates fork stabilization and restart and inhibits the cells from entering into mitosis, thus allowing the completion of DNA replication [102] (Figure 3).

Oncogene-induced RS not only contributes to cancer development by promoting genomic instability but it activates replication checkpoints, which slow down cell proliferation and trigger anti-cancer mechanisms leading to apoptosis or senescence [91,106,107,108]. Therefore, in cancer cells, RS is accurately regulated and a delicate equilibrium between RS occurrence and tolerance is achieved to sustain cancer progression.

To establish this subtle equilibrium, cells adapt to oncogene-induced RS avoiding severe replicative defects through a fine modulation of the ATR-CHK1 checkpoint response [109,110,111], E2 and ERα act as endogenous inhibitors of the ATR signaling cascade of the G2/M cell cycle checkpoint [112,113] (Figure 5). Indeed, the E2:ERα complex rapidly activates PI3K/AKT pathway and the resulting TOPBP1 phosphorylation reduces the DNA damage-dependent ATR:TOPBP1 association and the ATR kinase activity [112]. The E2:ERα-meditated AKT signaling also prevents the association of claspin and CHK1 leading to the inhibition of ATR-mediated CHK1 phosphorylation at Ser345 and promoting AKT-mediated CHK1 phosphorylation at Ser280. The latter event results in CHK1 sequestration in the cytoplasm and in the consequent overcoming of the checkpoint barrier also in the presence of RS [112].

In addition, their role in checkpoint signaling, clapsin, and TIM play also a role in the maintenance of replication fork integrity [50,114,115] increasing the resistance to RS and decreasing DDR signaling [116]. TIM expression in human BC positively correlates with ERα. Recently, TIM has been proposed as a novel key ERα interactor that enhances receptor transcriptional activity [117].

Therefore, it is not surprising that these factors are upregulated in many different types of cancer and that their overexpression is associated with a bad prognosis in BC [118,119,120]. TIM has been proposed as a molecular marker for predicting the response of ERα-positive postmenopausal BC to tamoxifen; moreover, TIM overexpression was associated with significantly shorter relapse-free survival [121].

### 5.3. Essential Functional Role of DDR and RSR Signaling in the Regulation of E2:ERα-Dependent Cell Proliferation

The available data demonstrate that E2 modulates in different manners the key regulators of the DDR and RSR pathways. E2 negatively modulates ATM and has protective effects against DNA damage and activates both DNA-PK and cyclin D1. On the contrary, E2 appears to inhibit the ATR-CHK1 signaling cascade via ERα, thus, reducing the activity of the RSR pathway (Figure 5).

The fact that E2 has a role both as a suppressor and as an inducer of the DDR and the RSR signaling implicates that the E2:ERα complex plays a critical role in balancing the proliferative and damaging stimuli induced by both the E2 chemical nature and the functional ERα activity. Consequently, the proteins regulating DDR and RSR pathways could provide selective proliferative advantages during BC progression.

#### DDR and RSR Pathways in ERα-Positive BC

To understand the impact of all the genes in the DDR and RSR pathways in BC cells, it is possible to inspect the publicly available CRISPR/CAS9 “dropout” screenings databases at the Broad and Welcome Sanger Institutes. These research centers evaluated the impact of all the genes encoded by the human genome on cell survival and proliferation in diverse cancer models including BC cells. These genome-wide loss-of-function screenings allow to define all the genes (and, therefore, pathways) essential for cancer cell proliferation and, in turn, to discover potential targets for cancer treatment (https://depmap.org/portal, https://score.depmap.sanger.ac.uk/, accessed on 29 March 2021; [122,123]). Salvati and co-workers systematically interrogated both datasets to identify molecular signatures corresponding to deregulated pathways enriched in ERα-positive BC [124]. Per each dataset, the authors considered the effect of the silencing of ~18,000 genes on the survival and proliferation of 11 ERα-positive BC cell lines and found 960 common essential genes. The subsequent functional annotation analysis revealed that those 960 common essential genes significantly enriched in 17 canonical pathways including those termed “estrogen receptor signaling”, “cell cycle control” and “assembly of RNA polymerase II complex” [124]. Moreover, their analysis revealed a critical functional role for “CHK proteins” and “G2/M DNA damage checkpoint” regulation in ERα-positive BC cell survival and proliferation [124]. Remarkably, 10 out of 17 pathways (i.e., 58,8%) contained at least one gene (e.g., ATR, CHK1, TOPBP1) involved in the control of genome integrity. This global view, together with the previously reported evidence, demonstrates that ERα-positive BC cells not only become addicted to the E2:ERα signaling, but also to the presence of the genes involved in the DDR and RSR pathways. 

These observations not only underscore a critical connection in the genes regulating the interplay among E2:ERα, DDR, and RSR pathways but also implicate DDR and RSR pathways in the E2:ERα-dependent control of cell proliferation and survival signaling in BC.

## 6. Discussion

The evidence summarized in this review indicates that E2 could act as a DNA damaging agent. Indeed, E2 can directly induce DNA damage causing genome instability via different mechanisms (i.e., E2-dependent metabolic by-products and E2:ERα activity) (Figure 5) [8,9,10,15]. Moreover, E2 down-regulates key effectors of DDR exacerbating its role as a DNA damaging inducer [10,112,125]. Although the E2:ERα complex can inhibit the activation of general repair pathways such as the ATR and/or ATM-related signaling, it can activate the DNA-PK pathway [100,112,126] (Figure 5). These contrasting results could be reconciled by considering the possibility that the redundant DDR pathways regulated by ATM, ATR, and DNA-PK could be part of the E2:ERα network. The strong waves of gene transcription and DNA synthesis occurring soon after E2 administration to BC cells [44,45] (Figure 2) could generate an RS (e.g., R-loops [15]). This stress affecting genome integrity could be resolved by the activation of the DDR and RSR pathways balancing, in a synchronized manner, the E2 transcriptional and replicative effects required for cell proliferation with the potential E2-dependent DNA damaging effects [16,100,127,128]. The potential detrimental effects of E2 during the hormone-induced physiological effects could be counteracted by the modulation of DDR pathways [124].

Although from the physiological point of view, it is very difficult to reconcile the E2-induced DNA damage with its well-known regulatory and beneficial effects, we propose a three-step model to explain the hypothesis formulated here (Figure 6).

(1) Under physiological conditions, the E2-induced DNA damaging effects caused by the activation of the E2:ERα signaling are buffered as mentioned above by the parallel ability of E2 to protect the cells from DNA damage (Figure 6, green). E2 plasma levels fluctuate in women in a range of concentrations between pM to nM [36]. Interestingly, E2 can work as a direct carcinogen both at physiological (i.e., <1 nM or lower) and supraphysiological (i.e., >1 µM) concentrations [85,86]. Therefore, it is tempting to speculate that E2 does not induce genotoxic effects only during the ovulatory phase (i.e., 48–72 h at 1 nM) in which the hormone exerts its physiological effects. From the evolutionary point of view, this assumption implies that rather than the chronic exposure to E2, the time of E2 administration is the critical parameter to achieve the maximal ERα functionality [37,43].

(2) In the initial phase of the E2-dependent cellular transformation (Figure 6, blue), the hormone and its metabolites induce DNA damage [4,5,6,8,129,130]. In addition, increased E2 plasma levels hyperactivate ERα-mediated transcriptional activity and DNA synthesis, thus further contributing to the initiation of breast carcinogenesis [37,40,41,42,43,44,45]. In this phase, besides the genes regulating cell cycle and cell proliferation, the resulting ERα-positive transformed cells overexpress DDR proteins (e.g., ATR and CHK1, BRCA1, TOBP1, and claspin) [124].

(3) Under pathological conditions (Figure 6, red) resulting from the transformed background described in the previous phase, the increased level of E2 in women with BC [4,5,6] is counteracted by a hyperactive E2-induced DNA repair activity [7,10,61,100]. In turn, this allows E2:ERα signaling to sustain BC tumor progression and spreading. In support of the model shown in Figure 6, it is important to recall that the treatment of BC consists in the inhibition of E2:ERα signaling either by reducing the circulating plasma E2 levels (i.e., ovariectomy, chemical castration, aromatase inhibitors treatment) or by inhibiting the ERα transcriptional activity [3,94].

## 7. Conclusions

DDR and RSR pathways are intrinsically connected with the activity of the E2:ERα signaling in BC cells and could be targeted to hamper BC cell proliferation. In this respect, it could be interesting to exploit the effect of specific inhibitors of the DDR kinases as novel drugs to be administered either alone/in combination with classic ET drugs (e.g., 4OH-Tam) or with novel compounds (e.g., CDK4/CDK6 inhibitors) used for the management of metastatic BC.

## Figures and Tables

**Figure 1 cancers-13-01583-f001:**
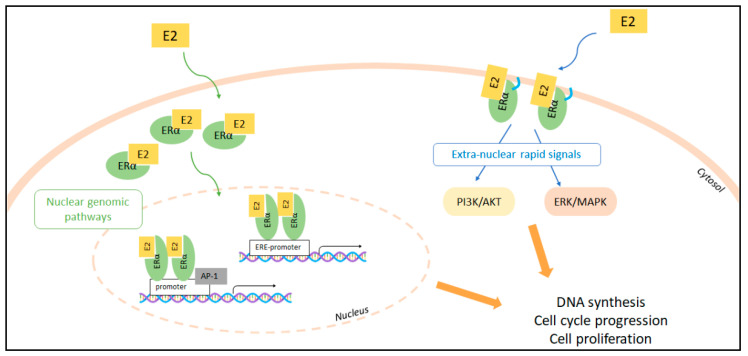
E2:ERα nuclear and extra-nuclear signaling to cell proliferation. The nuclear (left) and extra-nuclear (right) signaling pathways are indicated. Localization of the ERα to the cell plasma membrane by palmitoylation is indicated (blue sign). E2:ERα-dependent transcription of the genes containing or non-containing the estrogen response element (ERE) within their promoters as well as E2:ERα-dependent activation of the classic PI3K/AKT and ERK/MAPK kinase cascades are indicated. 17β-estradiol (E2); estrogen receptor α (ERα); Activator Protein 1 (AP-1); estrogen responsive element (ERE); phosphatidyl-inositol-3-kinase (PI3K); v-akt murine thymoma viral oncogene homolog 1 (AKT); mitogen-activated protein kinase (MAPK); extracellular regulated kinase (ERK).

**Figure 2 cancers-13-01583-f002:**
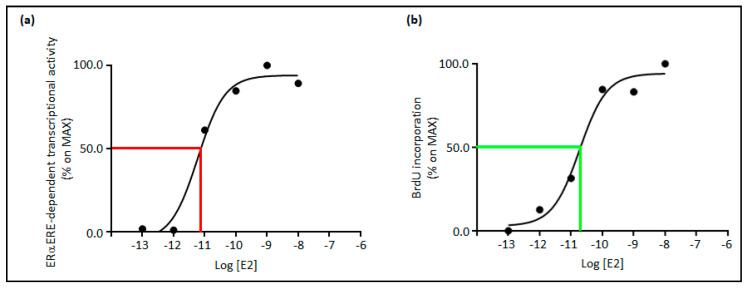
The E2 concentration-dependent effect on gene transcription and DNA synthesis. (**a**) ERα ERE-dependent transcriptional activity has been measured at 24 h in MCF-7 cells as described in [43] as a function of the indicated doses of 17β-estradiol (E2); (**b**) Bromodeoxyuridine incorporation (BrdU) in DNA has been measured at 24 h in MCF-7 cells as described in [43] as a function of the indicated doses of 17β-estradiol (E2). Red and green lines indicate half-maximal effect. For detail, see the text. 17β-estradiol (E2); estrogen receptor α (ERα); estrogen responsive element (ERE).

**Figure 3 cancers-13-01583-f003:**
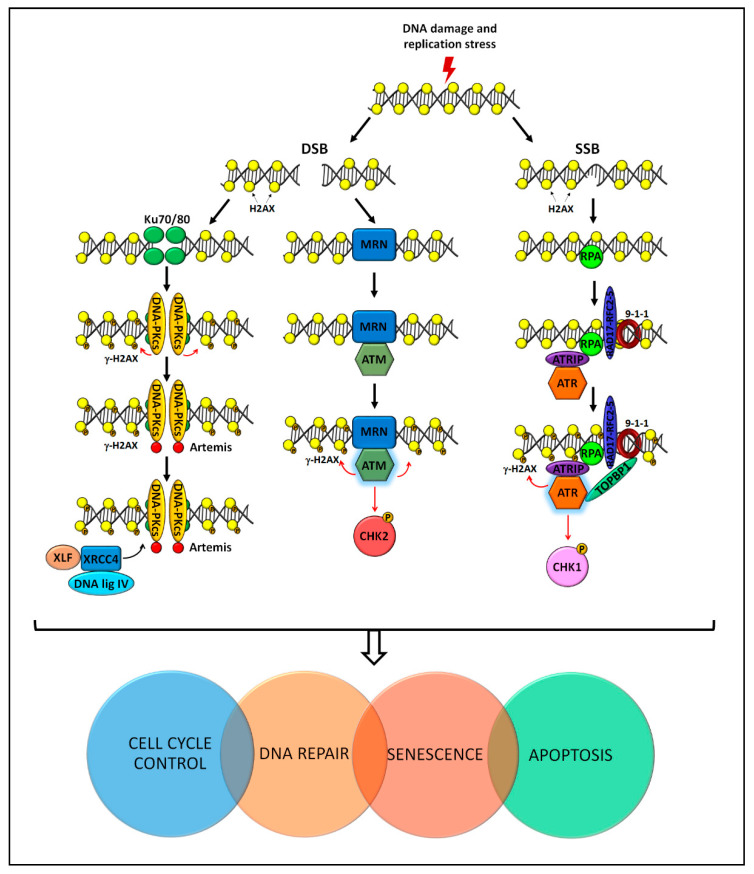
The DNA damage response (DDR) pathway involves sensor, transducer, mediator/amplifier, and effector proteins, which act following the induction of a DNA lesion or replication stress. The induction of single-strand breaks (SSB) or double-strand breaks (DSB) activates specific DDR kinases (i.e., DNA-PK, ATM, and ATR). In turn, this allows the activation of genes and proteins involved in cell cycle control and DNA repair. In the presence of highly damaged cells, apoptosis and senescence are activated to deplete or to semi-permanently arrest cells, respectively. For detail, see the text. X-Ray Repair Cross Complementing 6 (Ku70); X-Ray Repair Cross Complementing 5 (Ku80); catalytic subunit of DNA protein kinases (DNA-PKcs); phosphorylated H2A Histone Family Member X (γH2AX); Non-Homologous End Joining Factor 1 (XLF); X-Ray Repair Cross Complementing 4 (XRCC4); DNA ligase IV (DNA lig IV); MRE11/RAD50/NBS1 (MRN); Ataxia Telangiectasia Mutated (ATM); Checkpoint Kinase 2 (Chk2); Replication Protein A (RPA); Rad3-Related-Interacting Protein (ATRIP); DNA Topoisomerase II Binding Protein 1 (TOPBP1); Ataxia Telangiectasia and Rad3-Related Protein (ATR); Checkpoint Kinase 1 (Chk1).

**Figure 4 cancers-13-01583-f004:**
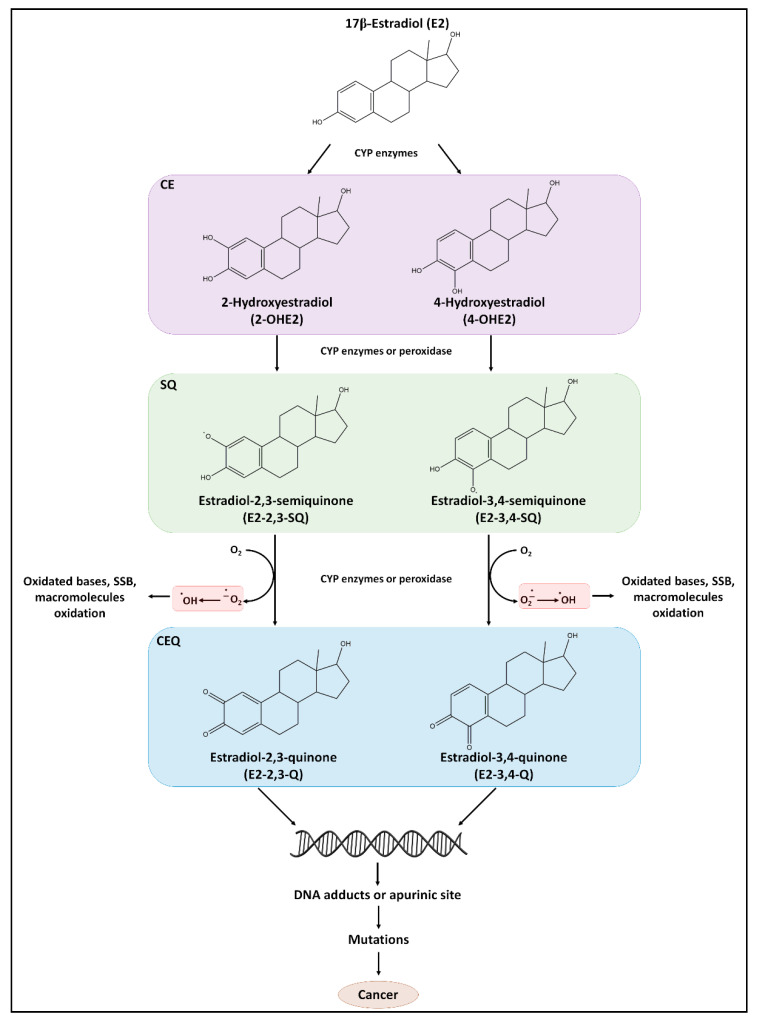
Metabolic byproducts of 17β-estradiol (E2) exerting genotoxic effects. 17β-estradiol (E2) is converted by cytochrome P450 (CYP)-dependent hydroxylation to 2-hydroxyestradiol (2-OHE2) and 4-hydroxyestradiol (4-OHE2). CYP enzymes or peroxidases convert these substrates into semiquinones (SQ), which may react with O_2_ to generate superoxide radicals (OH•) that are highly reactive and induces proteins and lipids oxidation, DNA bases oxidation, and single-strand breaks (SSB). SQ can be also converted to quinones (Q) that can covalently bind DNA thus inducing DNA and/or chromosomal damage. Overall, these events cause genome instability and cell transformation. For details, see the text.

**Figure 5 cancers-13-01583-f005:**
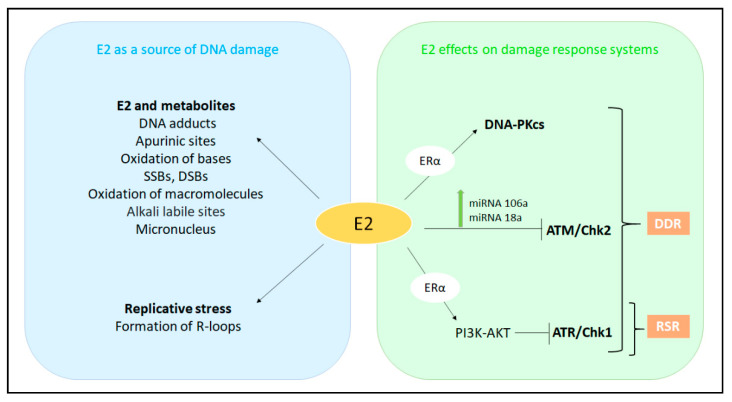
The interplay among E2, E2:ERα signaling and DNA damage, DNA damage response (DDR), and replication stress response (RSR). For details, see the text. 17β-estradiol (E2); estrogen receptor α (ERα); catalytic subunit of DNA protein kinases (DNA-PKcs); Ataxia Telangiectasia Mutated (ATM); Ataxia Telangiectasia and Rad3-Related Protein (ATR); Checkpoint Kinase 1 (Chk1); Checkpoint Kinase 2 (Chk2).

**Figure 6 cancers-13-01583-f006:**
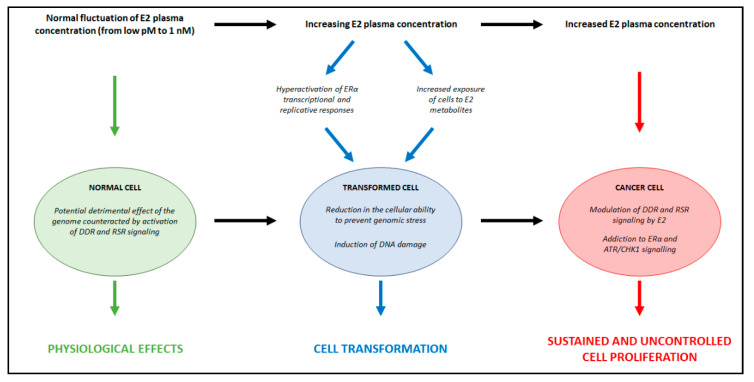
Proposed mechanism for the dual role of E2 in balancing DNA damage and genome integrity. Green, blue and red arrows indicate the effect in the specific cellular context. Black arrows indicate transitions between the specific cellular contexts. For details, see the text. 17β-estradiol (E2); estrogen receptor α (ERα); Ataxia Telangiectasia and Rad3-Related Protein (ATR); Checkpoint Kinase 1 (Chk1); DNA damage response (DDR); Replication stress response (RSR).

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
