# Peer review of "A Tale of Ice and Fire: The Dual Role for 17?-Estradiol in Balancing DNA Damage and Genome Integrity"

_cancers, 2021, doi:10.3390/cancers13071583_

Round 1

Reviewer 1 Report

Interesting review. The authors conclude the review by proposing that E2 could act as an aging agent, inducing DNA damage both through E2-dependent metabolic byproducts and because of the activity of the E2:ERα-complex. DDR, exacerbating its role as a DNA DSB. These conclusions derive from a careful analysis of the metabolic evidence and mechanisms of action dependent on E2 and the E2:Erα-complex. They conclude by pointing out that the DDR and RSR pathways are intrinsically connected with E2:ERα signaling activity in BC cells, and that molecules of these pathways could be molecular targets for stopping the progression of BC.

The evidence brought is mainly phenomenological. The pathways that mediate the effects of E2 on cancer cells still show gaps in knowledge and are still poorly understood.

One reason lies in the estrogen receptors themselves. The two different ER - ERα and ERβ - are encoded by two distinct human genes, ESR1 and ESR2, and the ERs are composed of functional domains, quite similar to other members of the nuclear receptor family. But by observing closely the chemical-physical and structural properties of these proteins, we discover that, despite the massive use of X-rays and NMR, their structures are partially resolved because they contain many large disordered segments in which post-translational modification sites are present (PTM). These segments are interspersed with nucleation centers of organized structure (mainly helicoidal) whose physical existence is fluctuating because it is time-dependent.

This means not only great flexibility of the physical structure of these proteins but also that they are present in solution with a wide distribution of equilibrium conformers (ensembles). The type of distribution and properties depend on the context and on the PTMs that generate many molecular-forms, each with its own characteristic functional property. A simple combinatorial algebra calculation shows how with three modifiable sites we can have 7 different molecular-forms, each of which is a single molecule with its specific chemical-physical and reactivity characteristics (ESR1 has 8 sites, ESR2 3 and there are many isoforms). Since the action of these molecules is expressed through their molecular-forms, in order to characterize a specific function in a cellular context at a precise metabolic moment, it is also necessary to know the molecular agent that induces PTM and the bio-molecule that interacts with the specific molecular form of the receptor to carry out the function. These unique events require an identical spacetime location for each actor. All this without considering that the encounter takes place in an extremely crowded cytoplasmic or nuclear environment that limits and distorts molecular diffusion and minimizes the likelihood of an encounter.

If we look at the BioGrid database, we see that for ESR1 it reports for humans the possibility of 2155 different physical interactions with as many proteins, while STRING, a database that reports functional interactions, shows for ESR1 220 different functional interactions that control 1258 biological processes (GO) and 216 molecular functions. For ESR2 we find fewer modifiable sites but still 2239 physical molecular-interactors and 43 different functional interactions that control 489 biological processes (GO) and 113 molecular functions. ISR1 has a net charge of + 5.3 at pH 7.0, and ISR2 + 13.0, values ​​far from those of globular proteins which are around 0 ± 1 at pH 7, charges that vary enormously with pH and PTMs. As a matter of fact, we have two strong polyampholytes or polyelectrolytes. If we check on SIGNOR, a database showing the functional interaction maps of the biomolecules, we find it involves these two proteins in at least 32 signaling pathways, which involve between 17 and 578 molecular actors. All this makes us understand how vast, complex, heterogeneous, and still unexplored the functional panorama of E2 and its two receptors is, but also that the molecular actors involved by the authors in their hypotheses are only a small part of the total actors and of the interconnected functions in which they are involved. The selection was, therefore, phenomenological because the spatiotemporal characteristics of each of them are unknown.

The review reports not only metabolic-functional glimpses involving these biomolecules, but the Authors also make hypotheses. They base them on their scientific expertise and the current state of knowledge. But I wonder, grafted into the correct context?

This referee suggests the Authors should report in a specific paragraph the context in which all this occurs by reporting structural, chemical-physical, interactomics, and functional characteristics of ISR1 and ISR2 in order to illustrate in which very complex context these molecules perform their function. This will make the review much more understandable for a wider audience that may not be very experienced in these issues.

Author Response

Reviewer #1

Interesting review. The authors conclude the review by proposing that E2 could act as an aging agent, inducing DNA damage both through E2-dependent metabolic byproducts and because of the activity of the E2:ERα-complex. DDR, exacerbating its role as a DNA DSB. These conclusions derive from a careful analysis of the metabolic evidence and mechanisms of action dependent on E2 and the E2:Erα-complex. They conclude by pointing out that the DDR and RSR pathways are intrinsically connected with E2:ERα signaling activity in BC cells, and that molecules of these pathways could be molecular targets for stopping the progression of BC.

The evidence brought is mainly phenomenological. The pathways that mediate the effects of E2 on cancer cells still show gaps in knowledge and are still poorly understood.

One reason lies in the estrogen receptors themselves. The two different ER - ERα and ERβ - are encoded by two distinct human genes, ESR1 and ESR2, and the ERs are composed of functional domains, quite similar to other members of the nuclear receptor family. But by observing closely the chemical-physical and structural properties of these proteins, we discover that, despite the massive use of X-rays and NMR, their structures are partially resolved because they contain many large disordered segments in which post-translational modification sites are present (PTM). These segments are interspersed with nucleation centers of organized structure (mainly helicoidal) whose physical existence is fluctuating because it is time-dependent.

This means not only great flexibility of the physical structure of these proteins but also that they are present in solution with a wide distribution of equilibrium conformers (ensembles). The type of distribution and properties depend on the context and on the PTMs that generate many molecular-forms, each with its own characteristic functional property. A simple combinatorial algebra calculation shows how with three modifiable sites we can have 7 different molecular-forms, each of which is a single molecule with its specific chemical-physical and reactivity characteristics (ESR1 has 8 sites, ESR2 3 and there are many isoforms). Since the action of these molecules is expressed through their molecular-forms, in order to characterize a specific function in a cellular context at a precise metabolic moment, it is also necessary to know the molecular agent that induces PTM and the bio-molecule that interacts with the specific molecular form of the receptor to carry out the function. These unique events require an identical spacetime location for each actor. All this without considering that the encounter takes place in an extremely crowded cytoplasmic or nuclear environment that limits and distorts molecular diffusion and minimizes the likelihood of an encounter.

If we look at the BioGrid database, we see that for ESR1 it reports for humans the possibility of 2155 different physical interactions with as many proteins, while STRING, a database that reports functional interactions, shows for ESR1 220 different functional interactions that control 1258 biological processes (GO) and 216 molecular functions. For ESR2 we find fewer modifiable sites but still 2239 physical molecular-interactors and 43 different functional interactions that control 489 biological processes (GO) and 113 molecular functions. ISR1 has a net charge of + 5.3 at pH 7.0, and ISR2 + 13.0, values ​​far from those of globular proteins which are around 0 ± 1 at pH 7, charges that vary enormously with pH and PTMs. As a matter of fact, we have two strong polyampholytes or polyelectrolytes. If we check on SIGNOR, a database showing the functional interaction maps of the biomolecules, we find it involves these two proteins in at least 32 signaling pathways, which involve between 17 and 578 molecular actors. All this makes us understand how vast, complex, heterogeneous, and still unexplored the functional panorama of E2 and its two receptors is, but also that the molecular actors involved by the authors in their hypotheses are only a small part of the total actors and of the interconnected functions in which they are involved. The selection was, therefore, phenomenological because the spatiotemporal characteristics of each of them are unknown.

The review reports not only metabolic-functional glimpses involving these biomolecules, but the Authors also make hypotheses. They base them on their scientific expertise and the current state of knowledge. But I wonder, grafted into the correct context?

This referee suggests the Authors should report in a specific paragraph the context in which all this occurs by reporting structural, chemical-physical, interactomics, and functional characteristics of ISR1 and ISR2 in order to illustrate in which very complex context these molecules perform their function. This will make the review much more understandable for a wider audience that may not be very experienced in these issues.

Author Response: The issue raised by this Reviewer is extremely interesting and refers to one of the main unanswered questions in the field of estrogen receptors that is the mechanistic relationship(s) among the physiological effects of E2, the basic biochemical and chemical-physical structural properties of the estrogen receptors, their functional cellular activities and the pleiotropic (and sometimes) contrasting phenotypic effects of the hormone in different cellular, tissue, and organs.

The Authors completely agree, as suggested by this Reviewer, that one possibility to understand this issue is the evaluation of all the possible structural conformations of the estrogen receptors determined by the fact that these proteins possess only two folded domains interspersed among long disordered peptide segments, which are themselves the target of a plethora of post-translational modifications, which further increase the number of possible structural protein conformations. This increases the possibility of the estrogen receptors interacting with many other proteins not only in a specific cellular context but also in a differential manner in diverse cellular contexts.

Therefore, understanding the context-specificity of the functioning of the estrogen receptors as a function of the above-mentioned structural conformations of the protein is paramount and it is a great suggestion. However, tackling these issues deserves another work (possibly an experimental one not a review as, to our knowledge no experimental works specifically addressed this question).

Indeed, we believe that discussing this point in the present review would lead the readers out of the scope of the work, which is to review the mechanisms by which E2 elicits DNA damage and protect cells from genomic instability as well as the ability of this hormone to directly control the signaling underlying DDR and RSR. Nonetheless, we agree with this Reviewer that adding a section clarifying that the E2 possesses context-specific effects (in this case the context is the mammary gland), which could depend on the repertoire of possibilities included in his/her comment, would help a non-expert reader to better appreciate the complexity of the topic. Therefore, we added it as a subsection in section 2.

Reviewer 2 Report

the main question addressed by the research is the estrogen capacity to elicit DNA damage by causing genome instability and to act as a direct carcinogen. In addition, the estrogen-mediated regulation of "DNA damage response" and "replicative stress response" signaling pathways is also reviewed. 

The issue reviewed here is both relevant and original as estrogens are well-known for their beneficial effects at the physiological level. Hence, the dual role of estrogens in balancing DNA damage and the genome integrity could be further deepened in order to set up innovative therapeutic strategies in breast cancer. 

The text is clear and easy to read and in general the paper is well written. The conclusions are consistent with the evidence and arguments presented and the Authors address the main question posed in the different sections of the manuscript.

Author Response

Reviewer #2

the main question addressed by the research is the estrogen capacity to elicit DNA damage by causing genome instability and to act as a direct carcinogen. In addition, the estrogen-mediated regulation of "DNA damage response" and "replicative stress response" signaling pathways is also reviewed. 

The issue reviewed here is both relevant and original as estrogens are well-known for their beneficial effects at the physiological level. Hence, the dual role of estrogens in balancing DNA damage and the genome integrity could be further deepened in order to set up innovative therapeutic strategies in breast cancer. 

The text is clear and easy to read and in general the paper is well written. The conclusions are consistent with the evidence and arguments presented and the Authors address the main question posed in the different sections of the manuscript.

Author Response: We thank this Reviewer for these comments.

Reviewer 3 Report

The Authors aimed to deal with the action of 17β-estradiol (E2) toward its role as homeostatic regulator of multifaceted functions and DNA damage agent. In the framework of these interesting issue, the Authors suggest that the action of E2 as a ‘genotoxic hormone’, which balances physiological and detrimental pathways to guarantee human body homeostasis, should be further considered.

The manuscript is well written and timely. My only concern is that the discussion/conclusions should be further extended in order to include the point of view of the Authors toward future directions aimed to better address the role of estrogens as genotoxic agents (i.e. the involvement of AHR and other factors/signaling pathways).

Some typos should be corrected.

Author Response

Reviewer #3

The Authors aimed to deal with the action of 17β-estradiol (E2) toward its role as homeostatic regulator of multifaceted functions and DNA damage agent. In the framework of these interesting issue, the Authors suggest that the action of E2 as a ‘genotoxic hormone’, which balances physiological and detrimental pathways to guarantee human body homeostasis, should be further considered.

The manuscript is well written and timely. My only concern is that the discussion/conclusions should be further extended in order to include the point of view of the Authors toward future directions aimed to better address the role of estrogens as genotoxic agents (i.e. the involvement of AHR and other factors/signaling pathways).

Some typos should be corrected.

Author Response: We thank this Reviewer for these comments. Regarding the future research perspectives, we believe that the most interesting aspect to further focus the attention on is the deep characterization of the impact of the main pathways involved in DDR and RSR signaling as potential drug targets for primary and metastatic breast cancer treatment. In particular, it will be interesting to exploit the effect of the DDR kinase inhibitors as potential new drugs for breast cancer treatment. We added this very last sentence in the conclusion section. We also have corrected the typos as per this Reviewer’s request.

Reviewer 4 Report

The Cancers review by Pescatori et al. (2020) provides a timely summary of the physiological and pathological functionality associated with circulatory estrogen.   The review provides a useful insight into the dichotomy of E2, and how concentration effects could induce the DNA damage associated with cellular transformation.  There are a few minor changes recommended to the manuscript prior to publication:

  1. Figure 1, parts of ERα (in two occasions) have been cut off during overlay of the label above it, can these be remedied? The authors use the terms MAPKs and ERK together, and PI3K and Akt, which is correct, but the way they are drawn in the figure, it could be misconstrued that the authors are insinuating that Akt and ERK lie further downstream of the corresponding PI3K and MAPK, respectively.  Signalling cascades are normally drawn with components closer to the membrane above more downstream targets.  The authors could draw MAPKs and ERKs together as a single entity and likewise PI3K/Akt to avoid confusion in the representation of the signalling cascade. For the Figure Legend and likewise the others that follow, the proteins represented by the abbreviations/acronyms should be included.

  1. Section 3 – the term dose is generally only used to describe an in vivo administered agent, not to describe an endogenous level/concentration. It is therefore suggested that the word dose is replaced with level or concentration in the title of this section, and in the sentences that follow, for example, these are concentration dependent effects, not technically dose-effects unless the E2 was administered to patients.

  1. Line 146 – CHIP should all be in capitals

  1. Could this section and those that follow that discuss the concentration effects of E2α result in the authors putting together either a graphical image or summary table that provides the reader with a useful, and quick reference to the endogenous concentration and associated biological effects?

  1. Figure 3 – as detailed above, for the Figure Legend, there is a need to ensure that all abbreviations and acronyms are included. For 02 the 2 should be subscripted.  The superoxide radical needs the word radical added after superoxide.

  1. ROS should be defined – need to ensure all acronyms are defined at first usage.

  1. Figure 4. PKCs the C should be capitalized.

  1. Within the text the word Fig. or fig has been used and even figure – need to be consistent and follow the journal style throughout.

  1. A suitable summation of the working hypothesis from the authors is presented as Figure 5. Not obligatory, but this figure could be improved aesthetically if incorporated all three stages, was within a cell, but moreover, if the authors could possibly include the (actual or indeed theoretical) E2 concentration ranges associated with each of the stages, that would be useful to the reader, although this may not be possible given the detail included in Discussion section 1.

  1. Typos and grammatical check needed.

Author Response

Reviewer #4

The Cancers review by Pescatori et al. (2020) provides a timely summary of the physiological and pathological functionality associated with circulatory estrogen.   The review provides a useful insight into the dichotomy of E2, and how concentration effects could induce the DNA damage associated with cellular transformation.  There are a few minor changes recommended to the manuscript prior to publication:

  1. Figure 1, parts of ERα (in two occasions) have been cut off during overlay of the label above it, can these be remedied? The authors use the terms MAPKs and ERK together, and PI3K and Akt, which is correct, but the way they are drawn in the figure, it could be misconstrued that the authors are insinuating that Akt and ERK lie further downstream of the corresponding PI3K and MAPK, respectively.  Signalling cascades are normally drawn with components closer to the membrane above more downstream targets.  The authors could draw MAPKs and ERKs together as a single entity and likewise PI3K/Akt to avoid confusion in the representation of the signalling cascade. For the Figure Legend and likewise the others that follow, the proteins represented by the abbreviations/acronyms should be included.

Author Response: We thank this Reviewer for noticing these inaccuracies. We have now corrected them as per this request.

  1. Section 3 – the term dose is generally only used to describe an in vivo administered agent, not to describe an endogenous level/concentration. It is therefore suggested that the word dose is replaced with level or concentration in the title of this section, and in the sentences that follow, for example, these are concentration dependent effects, not technically dose-effects unless the E2 was administered to patients.

Author Response: We thank this Reviewer for noticing these inaccuracies. We have now corrected them as per this request.

  1. Line 146 – CHIP should all be in capitals

Author Response: We have put CHIP all in capitals.

  1. Could this section and those that follow that discuss the concentration effects of E2α result in the authors putting together either a graphical image or summary table that provides the reader with a useful, and quick reference to the endogenous concentration and associated biological effects?

Author Response: We have added a figure depicting the concentration-dependent effects of E2 on DNA synthesis and ERE-based transcriptional activity in breast cancer cells according to this request and as per the description in the text.

  1. Figure 3 – as detailed above, for the Figure Legend, there is a need to ensure that all abbreviations and acronyms are included. For 02 the 2 should be subscripted.  The superoxide radical needs the word radical added after superoxide.

Author Response: We have corrected the figure caption as per this request.

  1. ROS should be defined – need to ensure all acronyms are defined at first usage.

Author Response: We have defined ROS and checked that the acronyms are defined at the first usage as per this request.

  1. Figure 4. PKCs the C should be capitalized.

Author Response: We have corrected the figure as per this request.

  1. Within the text the word Fig. or fig has been used and even figure – need to be consistent and follow the journal style throughout.

Author Response: We have corrected these inaccuracies as per this request.

  1. A suitable summation of the working hypothesis from the authors is presented as Figure 5. Not obligatory, but this figure could be improved aesthetically if incorporated all three stages, was within a cell, but moreover, if the authors could possibly include the (actual or indeed theoretical) E2 concentration ranges associated with each of the stages, that would be useful to the reader, although this may not be possible given the detail included in Discussion section 1.

Author Response: We have modified the figure as per this request.

  1. Typos and grammatical check needed.

Author Response: We have carefully checked the text for typos.

Round 2

Reviewer 1 Report

I thank the Authors for their consideration. New technologies (e.g., new single-cell sequencing and transcriptomics) allow us to study the molecular mechanisms underlying the functions of many protein complexes. They clarify where, when, and in what form proteins act. However, many labs produce data with thirty-year-old ideas, therefore, it is difficult to make them understand the frontier has shifted.

Your review is a good job, and I suggest its publication.